# Gynecological Surveillance and Surgery Outcomes in Dutch Lynch Syndrome Carriers

**DOI:** 10.3390/cancers13030459

**Published:** 2021-01-26

**Authors:** Ellis L. Eikenboom, Helena C. van Doorn, Winand N. M. Dinjens, Hendrikus J. Dubbink, Willemina R. R. Geurts-Giele, Manon C. W. Spaander, Carli M. J. Tops, Anja Wagner, Anne Goverde

**Affiliations:** 1Department of Clinical Genetics, Erasmus MC Cancer Institute, University Medical Center Rotterdam, 3000 CA Rotterdam, The Netherlands; e.eikenboom@erasmusmc.nl (E.L.E.); a.wagner@erasmusmc.nl (A.W.); 2Department of Gastroenterology and Hepatology, Erasmus MC Cancer Institute, University Medical Center Rotterdam, 3000 CA Rotterdam, The Netherlands; v.spaander@erasmusmc.nl; 3Department of Gynecology, Erasmus MC Cancer Institute, University Medical Center Rotterdam, 3000 CA Rotterdam, The Netherlands; h.vandoorn@erasmusmc.nl; 4Department of Pathology, Erasmus MC Cancer Institute, University Medical Center Rotterdam, 3000 CA Rotterdam, The Netherlands; w.dinjens@erasmusmc.nl (W.N.M.D.); h.dubbink@erasmusmc.nl (H.J.D.); w.geurts-giele@erasmusmc.nl (W.R.R.G.-G.); 5Department of Clinical Genetics, Leiden University Medical Center, 2333 ZA Leiden, The Netherlands; C.M.Tops@lumc.nl

**Keywords:** Lynch syndrome, gynecological surveillance, endometrial cancer, ovarian cancer, DNA mismatch repair, risk-reducing surgery, *MLH1*, *MSH2*, *MSH6*, *PMS2*

## Abstract

**Simple Summary:**

Female Lynch syndrome (LS) carriers have an increased risk to develop endometrial and ovarian cancer. In the Netherlands, carriers are therefore advised annual gynecological surveillance and eventually, risk-reducing surgery. Global gynecological LS surveillance guidelines are scarce and based on limited evidence. These are, however, warranted to offer accurate surveillance. To provide more insight into surveillance outcomes, this study assessed outcomes of gynecological surveillance and risk-reducing surgery in 164 LS carriers diagnosed in our center, with a median follow-up of 5.6 years per carrier. Although most surveillance visits happened within an advised timeframe, we observed large variability in how gynecological surveillance visits were performed. This finding stresses the need for development of clear and evidence-based guidelines. Endometrial cancers identified at surveillance were all found in early stage, mostly symptomatic, questioning surveillance benefit. Large, prospective studies should assess to what extent current LS surveillance programs contribute to early detection of gynecological tumors.

**Abstract:**

Lynch syndrome (LS) is caused by pathogenic germline variants in DNA mismatch repair (MMR) genes, predisposing female carriers for endometrial cancer (EC) and ovarian cancer (OC). Since gynecological LS surveillance guidelines are based on little evidence, we assessed its outcomes. Data regarding gynecological tumors, surveillance, and (risk-reducing) surgery were collected from female LS carriers diagnosed in our center since 1993. Of 505 female carriers, 104 had a gynecological malignancy prior to genetic LS diagnosis. Of 264 carriers eligible for gynecological management, 164 carriers gave informed consent and had available surveillance data: 38 *MLH1*, 25 *MSH2*, 82 *MSH6*, and 19 *PMS2* carriers (median follow-up 5.6 years). Surveillance intervals were within advised time in >80%. Transvaginal ultrasound, endometrial sampling, and CA125 measurements were performed in 76.8%, 35.9%, and 40.6%, respectively. Four symptomatic ECs, one symptomatic OC, and one asymptomatic EC were diagnosed. Endometrial hyperplasia was found in eight carriers, of whom three were symptomatic. Risk-reducing surgery was performed in 73 (45.5%) carriers (median age 51 years), revealing two asymptomatic ECs. All ECs were diagnosed in FIGO I. Gynecological management in LS carriers varied largely, stressing the need for uniform, evidence-based guidelines. Most ECs presented early and symptomatically, questioning the surveillance benefit in its current form.

## 1. Introduction

Lynch syndrome (LS) is one of the most prevalent hereditary cancer predisposition syndromes [1]. LS is caused by germline mutations in DNA mismatch repair (MMR) genes *MLH1*, *MSH2*, *MSH6*, *PMS2,* or by a deletion of the 3′ end of the *EpCAM* (*TACSTD1*) gene. Pathogenic germline variants in MMR genes, causative for LS, were first discovered in the 1990s, with *MSH2* being first (1993), followed by *MLH1* (1994) and *PMS2* (1994), *MSH6 (1997)*, and *EpCAM* (2009) [2,3,4,5,6]. Besides an increased risk of up to 57.1% to develop colorectal cancer (CRC), female carriers are also at high risk to develop gynecological malignancies, such as endometrial cancer (EC) and cancer in the ovaries (OC) [7,8,9]. A recent study showed that the risks for EC development by age 75 are 37%, 48%, 41%, and 12% for *MLH1*, *MSH2*, *MSH6,* and *PMS2* carriers, respectively. The risk of developing OC by age 75 was found to be between 3% and 17%, depending on MMR gene involved [10]. Besides CRC, EC, and OC, LS carriers are also at an increased risk to develop other types of tumors, such as tumors of the small intestine, brain, and skin. Although LS carriers have an increased risk, compared to the general population, to develop these tumors, these risks are lower than for CRC, EC, and OC development [11].

Development of surveillance strategies, such as biennial colonoscopy, led to a decrease in CRC-related death [12,13,14]. Female LS carriers are globally also advised a regular gynecological examination [15,16]. Exact guidelines differ per country, but usually include at least annual or biennial gynecological examination with transvaginal ultrasound to assess the endometrium and ovaries in women 25–40 years to 60 years [17,18,19,20,21]. Uniform guidelines for gynecological surveillance are not yet available: on the one hand, little is known about premalignant stages of LS-associated gynecological tumors. On the other hand, it is hitherto not clear to what extent screening can contribute to the detection of gynecological tumors in early stages [15,21,22,23,24]. In the Netherlands, guidelines regarding gynecological management in LS carriers were revised in December 2015. Before this time, LS carriers could opt for annual to biennial gynecological assessment, from the age of 30 to 35 years. From 2016 onwards, female LS carriers are advised to have an annual transvaginal ultrasound, endometrial sampling, and assessment of the ovaries by ultrasound and/or CA125 measurements between age 40 and 60 years [19]. The age 40–60 years is based on the assumption that EC symptoms, such as post-menopausal bleeding, could potentially be misinterpreted as perimenopausal bleeding. In case carriers have a relative with a gynecological tumor identified before the age of 40 years, carriers are advised to start surveillance five years before this age. From the initiation of gynecological surveillance programs, carriers have been extensively informed about (early) symptoms of a gynecological tumor, regardless of their age, as the majority of ECs presents with gynecological complaints, such as irregular bleeding [25]. Possibilities of risk-reducing surgery are also being continuously discussed, such as a total hysterectomy and/or bilateral salpingo-oophorectomy after completed childbearing, to further decrease the risk of gynecological cancer development [26]. Nevertheless, little is known about the yield of gynecological surveillance in LS carriers. Additionally, the outcomes of the Dutch gynecological surveillance strategy have never been assessed so far. However, understanding these outcomes may be useful for the development of tailored gynecological management strategies for these carriers. Therefore, the results of gynecological management in female LS carriers diagnosed at our center were assessed.

## 2. Materials and Methods

For this retrospective cohort study, we assessed prevalence of gynecological tumors, outcomes of surveillance and the yield of risk-reducing surgery in female carriers of a pathogenic variant in one of the MMR genes, from here on referred to as female LS carriers, diagnosed in our center. Outcomes were assessed for the different MMR gene carrier groups. As *EpCAM* carriers were previously found to have a significantly different risk to develop EC compared to *MSH2* carriers [27], these carriers were excluded from further analyses. We did not include carriers with biallelic pathogenic MMR germline variants (congenital mismatch repair deficiency) or LS carriers with pathogenic variants in more than one MMR gene. Gynecological surveillance was not applicable in case of post-mortem LS diagnosis, in carriers having had gynecological surgery prior to LS diagnosis or in case carriers were aged younger or older than the advised age for surveillance. From cases with gynecological surgery due to EC or OC, age at diagnosis and the 2009 Fédération Internationale de Gynécologie et d’Obstétrique (FIGO) stage for the classification of gynecological tumors [28] were assessed. 

### 2.1. Patient Selection and Data Extraction

The department of Clinical Genetics of the Erasmus MC, University Medical Center in Rotterdam, serves as a regional referral center for the south-west of the Netherlands. Upon LS diagnosis, a genetic counselor informs LS carriers about the gynecological tumor risk and the possibilities of surveillance and/or surgery to reduce these risks. LS carriers who are (nearly) eligible for gynecological surveillance or surgery are referred to the department of gynecology. Carriers referred to the department of gynecology in our center receive a brochure summarizing these features.

From our database, we selected all female LS carriers diagnosed between 1993 and 2020. Subsequently, female LS carriers eligible for gynecological surveillance and/or risk-reducing surgery were asked for informed consent to retrieve and analyze their data regarding gynecological surveillance and surgery. Data were retrieved from the hospital charts where surveillance or surgery had been performed. Data on surveillance visits included the number of surveillance visits, interval between two subsequent visits, outcome of transvaginal ultrasound, endometrial sampling, and Cancer Antigen 125 (CA125) measurements. CA125 is a tumor marker, and may be elevated in the blood of patients with OC and EC [29]. Data on risk-reducing surgery included type and date of surgery, pathology, and development of gynecological tumors. Data were retrieved up to October 2020. In case patient charts did not mention any information whatsoever regarding transvaginal ultrasound, CA125 measurements, or endometrial sampling, we assumed the corresponding assay was not performed. If the endometrial thickness was >10 mm in premenopausal women during the proliferative phase or >4 mm in postmenopausal women, or if the medical report mentioned a “thickened” endometrium, we concluded that the endometrial thickness was abnormal. CA125 values were considered to be elevated when >35 kU/L. All surveillance visits before risk-reducing surgery were included in our study. Permission of the Erasmus Medical Center Committee on Research Involving Human Subjects was granted (MEC-2020-0600).

### 2.2. Statistical Analyses

Data were analyzed in SPSS statistical software version 25.0. Differences in baseline characteristics were assessed by a χ^2^ test (or Fisher’s exact test in the case of paucity of data) or Kruskal–Wallis test, for categorical and quantitative variables, respectively. *p*-values < 0.05 were considered statistically significant.

## 3. Results

### 3.1. Patient Characteristics 

Since 1993, we diagnosed a pathogenic MMR germline variant in 505 females (*MLH1 n* = 107, *MSH2 n* = 90, *MSH6 n* = 224, and *PMS2 n* = 84). In 241 carriers, gynecological surveillance and surgery was not applicable (Figure 1). In 104 of these 241 carriers, the indication for surgery had been a gynecological malignancy (EC *n* = 85, OC *n* = 17, and gynecological organ of origin not specified *n* = 2). Forty-one ECs (80.4%) were found in FIGO stage I, three in FIGO stage II (5.9%), three in FIGO stage III (5.9%), four in FIGO stage IV (7.8%), and unknown in 34 ECs. The majority of the ECs were identified in age category 40–60 years (*n* = 60, 71.4%); two ECs in patients younger than 40 years (2.4%), 21 in patients older than 60 years (25.0%), and for one patient, age at EC diagnosis was unknown. Ten OCs (58.8%) were found in FIGO stage I, two in FIGO stage II (11.8%), three in FIGO stage III (17.6%), one in FIGO stage IV (5.9%), and unknown in one. The majority of the OCs were diagnosed in age category 40–60 years (*n* = 9, 52.9%); four OCs were diagnosed in patients younger than 40 years of age (23.5%), three in patients older than 60 years of age (17.6%), and for one patient, age at OC diagnosis was unknown.

Of the 264 carriers eligible for gynecological surveillance and/or risk-reducing surgery, gynecological management data were available for 164. Data were not available because carriers died before the time of asking informed consent, because they did not grant informed consent to retrieve their medical data, or because carriers had not been enrolled in a gynecological surveillance program. Of the 164 carriers with gynecological surveillance data, 38 (23.2%) carried a pathogenic germline variant in *MLH1*, 25 (15.2%) in *MSH2*, 82 (50.0%) in *MSH6*, and 19 (11.6%) in *PMS2* (Table 1). The mean ages at first gynecological surveillance visit before 2016 and from 2016 onwards were 46.0 years (interquartile range (IQR) 37.9–53.6 years) and 53.8 years (IQR 42.4–61.1), respectively, and did not differ significantly between *MLH1*, *MSH2*, *MSH6*, and *PMS2* carrier groups. As expected due to adjustment of the Dutch LS gynecological surveillance guidelines in December 2015, LS carriers were overall significantly older at the first surveillance visit from 2016 onwards (*p* = 0.007).

### 3.2. Surveillance

#### 3.2.1. Characteristics of Surveillance

On average, the 164 LS carriers under surveillance had 3 (IQR 2–6) surveillance visits, corresponding with a median of 5.6 follow-up years (IQR 3.0–9.0 years) per carrier. Median number of follow-up years was significantly higher for *MLH1*/*MSH2* carriers than for *MSH6*/*PMS2* carriers (*p* = 0.009, Table 1). 

Transvaginal ultrasounds, endometrial sampling, and CA125 measurements were performed in 76.8%, 35.9%, and 40.6% of all surveillance visits, respectively. In 60.8% of carriers, at least one CA125 measurement was performed. In gynecological surveillance visits of *MLH1* and *MSH6* carriers, CA125 measurements were significantly more often performed, compared to those of *MSH2* and *PMS2* carriers (54.9% and 44.0%, vs. 16.5% and 18.1%, respectively; *p* < 0.001). Endometrial sampling was significantly more often performed in *PMS2* carriers (*p* < 0.001). In all LS carriers under 50 years of age, endometrial sampling was significantly more often performed compared to carriers over 50 years of age (*p* = 0.024).

#### 3.2.2. First Surveillance Visit

In 57 of 164 carriers (34.8%) data of only one surveillance visit were available (Figure 1, Table 2). Of all carriers, *MSH6* carriers had significantly more often just one surveillance visit (*p* = 0.040). Reasons for cessation of the follow-up varied, with the majority of carriers (*n* = 30) having subsequent risk-reducing surgery. Upon the first surveillance visit, one EC was identified in a 64-year old *MSH6* carrier, without gynecological complaints (Table 2 and Table 3). Ten of the 164 carriers (6.1%) had another abnormality identified upon the first surveillance visit: an increased CA125 value (*n* = 2), a thickened endometrium (*n* = 4), and endometrial hyperplasia (*n* = 5). All the latter carriers opted for subsequent risk-reducing surgery; pathological assessment did not show the presence of an EC in four, a pathology report was not available in one.

#### 3.2.3. Second or Higher Surveillance Visit

For 107 carriers, the data of subsequent surveillance visits were available. The median interval between subsequent surveillance visits was 1.0 years (before 2016 IQR 0.9–1.4 years, from 2016 onwards IQR 1.0–1.2, Table 2). Before 2016, *MSH2* and *PMS2* carriers had a significantly longer interval between subsequent surveillance visits (1.1 years, IQR 1.0–2.0 for *MSH2*, IQR 1.0–1.4 for *PMS2*, *p* < 0.001). The median interval in these carriers decreased from 2016 onwards, having no significant difference between carrier groups (*p* = 0.277). In carriers enrolled in surveillance before 2016, who were advised to have gynecological surveillance every 1–2 years, a median of 82.3% of subsequent surveillance visits was performed within 24 months. Carriers with surveillance visits from 2016 onwards were advised annual gynecological surveillance: here, a median of 83.3% of subsequent surveillance visits was performed within 15 months.

After the first surveillance visit, five ECs were identified (two *MLH1* carriers, two *MSH2*, and one *MSH6*; Table 2 and Table 3); all were found in FIGO stage I. One of these ECs was found by endometrial biopsy in carriers without gynecological complaints, the remaining four were found in carriers with gynecological complaints. Upon pathological assessment after risk-reducing surgery, another ECs was found in a 58-year old *MSH6* carrier (Table 3 and Table 4). In five of these six carriers with EC, the tumor was identified at age 40–60 years (Table 3). Of note, in only two patients diagnosed with EC during surveillance, both transvaginal ultrasound and endometrial sampling were performed at the previous gynecological surveillance visit and these assays did not show clues for a premalignancy. One FIGO stage IV OC was diagnosed in a 48-year old *MSH2* carrier, who was presented with blood loss. At her previous visit less than one year earlier, the transvaginal ultrasound was unremarkable (Table 2 and Table 3). 

#### 3.2.4. Abnormalities Indicative for (pre)Malignancy

Other abnormal tests were found in 19 carriers upon second or higher surveillance visits: an increased CA125 value (*n* = 6), a thickened endometrium (*n* = 10), and hyperplasia (*n* = 3, Table 2). The latter three carriers opted for gynecological surgery, but pathological assessment did not result in identification of an EC. Additionally, endometrial sampling was suggestive for premalignancy for an *MLH1* carrier. Subsequent hysterectomy with bilateral salpingo-oophorectomy resulted in identification of an EC (Table 3).

### 3.3. Risk-Reducing Surgery

Of all 164 carriers enrolled in gynecological surveillance, 73 (45.5%) underwent gynecological surgery (Table 4). In 53 carriers, surgery was performed as a risk-reducing measure, whereas in 13 carriers, surgery was performed due to suspicion of (pre)malignancies. In three cases, surgery was performed for other gynecological problems, like uterine myomas, and for the remaining five cases, the reason for gynecological surgery was unknown. The procedures performed were hysterectomy with bilateral salpingo-oophorectomy (*n* = 58), hysterectomy (*n* = 11), and ovariectomy with or without removal of the fallopian tubes (*n* = 4, Table 4). Reasons for the performance of ovariectomy without simultaneous hysterectomy, for example as risk-reducing surgery or other reasons, could not be retrieved for these carriers. Pathological assessment revealed one EC in a 58-year old *MSH6* carrier, without gynecological complaints (Table 3 and Table 4). 

In general, surgery was performed at a median age of 51 years (IQR 45–55 years before 2016 and IQR 47–56 since 2016), with no significant age differences between carrier groups. In general, three surveillance visits (IQR 1–5) preceded this surgery. *MSH6* carriers opted significantly more for gynecological surgery compared to the other MMR gene carrier groups (*n* = 44, 53.7% of all *MSH6* carriers enrolled, *p* = 0.003; Table 4).

## 4. Discussion

In the current study, we assessed gynecological tumors, surgery, and surveillance outcomes in 505 LS carriers. Of these carriers, 104 had a gynecological malignancy prior to their genetic diagnosis of LS. Outcomes of gynecological management were assessed in 164 LS carriers, eligible for gynecological surveillance and/or risk-reducing surgery. More than 80% of surveillance visits were performed within the advised timeframe. Endometrial sampling revealed endometrial hyperplasia in eight carriers, of whom three symptomatic carriers. Additionally, endometrial sampling revealed an EC in two asymptomatic carriers. Additionally, four ECs and one OC were found in carriers with gynecological complaints during the study period. Almost half of the carriers under surveillance (45.5%), and particularly *MSH6* carriers, opted for risk-reducing surgery, at a median age of 51 years. Pathological assessment of the uterus after risk-reducing surgery revealed one EC in an asymptomatic *MSH6* carrier. To our knowledge, this is the first study with a relatively large cohort of especially *MSH6* carriers examining the outcomes of gynecological management guidelines in the Netherlands.

### 4.1. Surveillance 

In the Netherlands, there is a relatively high prevalence of *MSH6* pathogenic variants. Some Dutch studies particularly indicated a high EC risk for these carriers [30,31]. This has probably influenced risk counseling in female *MSH6* carriers. A more stringent risk-reducing surgery advice for these carriers could therefore partially explain the significant longer median follow-up of *MLH1* and *MSH2* carriers, compared to *MSH6* and *PMS2* carriers in our cohort. Additionally, genetic testing for *MSH2* and *MLH1* was available earlier.

Although efficacy of gynecological screening is being questioned by some [23], two studies performed in the Netherlands did show that premalignant lesions and early EC could be identified in LS carriers [32,33]. In our cohort, transvaginal ultrasound was performed in about three quarters of surveillance visits. A recent study showed that its application generally causes little discomfort in post-menopausal women [34]. Addition of endometrial sampling was found to be more efficient in EC diagnosis than transvaginal ultrasound alone [18,33,35,36]. In our cohort, endometrial sampling was performed in about one third of visits. This could potentially be attributed to its painful application [37]. It can be hypothesized that, particularly in an asymptomatic postmenopausal woman with a thin endometrial thickness, the posttest chance of finding an EC is so small that the patient and her physician decide that sampling does not weigh in. CA125 measurements, useful in identification of OC and also advised by national Dutch surveillance guidelines, were performed in 40% of surveillance visits. Some hospitals may only perform CA125 measurements at the first surveillance visit and not routinely at every subsequent visit. However, as CA125 measurements were performed at least once in slightly more than 60% of carriers, this might need attention. Reason for the variation in application of transvaginal ultrasound, endometrial sampling, and CA125 measurements could potentially be found in the fact that surveillance takes place in different hospitals. A recent study by Ryan et al. showed regional variation in gynecological surveillance visits. Additionally, about one-third of the gynecological oncologists were unfamiliar with gynecological surveillance guidelines [38].

### 4.2. Tumors 

Among the studied carriers eligible for surveillance, hyperplasia was identified in eight carriers, of whom three had gynecological complaints. Four ECs and one OC were found in carriers having gynecological complaints. Endometrial biopsy revealed two ECs in asymptomatic carriers. One of these carriers had endometrial sampling performed in her visit prior to the identification of the tumor, without clues for premalignancy. One EC was diagnosed upon pathological assessment after risk-reducing surgery, in an asymptomatic carrier. In our study, three carriers with endometrial hyperplasia and more than half of carriers with EC were symptomatic. On the one hand, these findings question whether, and if so, to what extent, female LS carriers may benefit from gynecological surveillance programs. On the other hand, these findings also highlight the importance of informing LS carriers about gynecological complaints, indicative for an underlying (pre)malignancy. Furthermore, these findings stress the importance of gynecological assessment in the case of gynecological complaints, regardless of age, as blood loss in a 37-year old carrier led to the identification of an EC.

### 4.3. Difference in Risk-Reducing Surgery

Risk-reducing surgery, usually hysterectomy combined with bilateral salpingo-oophorectomy, was shown to be both effective and cost-effective in decreasing the gynecological cancer risk [26,39]. In our cohort, about half of patients opted for risk-reducing surgery, of which more than three-quarters chose for hysterectomy combined with bilateral salpingo-oophorectomy. *MSH6* carriers chose significantly more often for risk-reducing surgery. This was in contrast to *MSH2* carriers (32.0% vs. 53.7% in *MSH6* carriers), whereas these carriers were recently predicted as having the highest EC risk of all carrier groups [10]. Notably, in 18 *MSH2* carriers with follow-up data available, two ECs and one endometrial hyperplasia were diagnosed in our study, representing the highest amount of (pre)malignancies of all carrier groups. The fact that *MSH2* carriers chose less often for risk-reducing measures, could potentially be due to a phenomenon also noticed by Sun et al.: in this study, female LS carriers with a family history of cancer chose surprisingly less often for risk-reducing measures than their counterparts without a family history of cancer [40]. The authors attribute this to possible favorable experiences of relatives with endometrial cancer, since symptoms allow early diagnosis of the cancer in a curable stage. In case *MSH2* carriers in our study did opt for risk-reducing surgery, they were younger.

### 4.4. Relevance

Although it has not yet been proven that gynecological surveillance causes identification of a gynecological tumor in an earlier stage, guidelines currently advise LS carriers regular examination [15]. Surveillance is advised in view of the high risk of developing gynecological tumors, which can be as high as 48% [10]. Additionally, EC is often the presenting tumor in female LS carriers [41]. Assessment of gynecological management strategies is thus of utmost importance and functions as a first step in the quest for more tailored surveillance guidelines. Several factors hamper the development of these guidelines: first, in contrast to CRC, little is known about the precursor lesions preceding EC or OC, especially in LS carriers. Some studies, however, showed that mismatch repair deficiency already occurs early in the carcinogenesis pathway [22,42]. Second, identification of precursor lesions in practice is hindered in premenopausal women, as endometrial sampling was shown to have lower specificity and sensitivity in these women [43]. Third, it is not certain if surveillance-detected cancers differ from incidence cancers: EC often presents with vaginal bleeding, particularly in postmenopausal women. Consequently, ECs are mostly detected in early stage with a favorable outcome. Therefore, some advocate that gynecological surveillance in LS carriers should mainly take place based on complaints. A previous study, that assessed OCs in LS carriers, found OCs in a predominantly early stage, but noticed that annual gynecological surveillance was not causative for identification in the early stage [44]. In the current study, however, one OC was identified in FIGO stage IV, despite gynecological complaints.

### 4.5. Strengths and Limitations 

Strengths of our study were the relatively large cohort of LS carriers, which made it possible to mutually compare the four LS carrier groups. Second, the acquisition of extensive data among others risk-reducing surgery and assays performed during these visits, provided us with an accurate overview of gynecological management in these carriers. 

Our study had several limitations. First, due to the retrospective nature of this study, we were hampered by missing data. Therefore, for some carriers, some surveillance visits might not have been included in our analysis and non-surveillance visits might be misinterpreted. Bias could be introduced because we had no information on the women who did not grant informed consent to retrieve their gynecological management data and women who did not have screening after LS diagnosis. To what extent these carriers were enrolled in a gynecological management program remained unknown. As a consequence, development of EC in both groups could not be compared. Second, although our cohort included a relatively large number of LS carriers when compared to previous literature, larger groups are needed to draw definite conclusions about surveillance efficacy. Our cohort consists of patients included from 1993 to 2020. During this time span, several changes can be noted: the recognition of new mutations, changes in cancer incidence data, and screening protocols having evolved over time. This, however, is an ongoing process: today’s recommendations will be adjusted in the future. To accurately assess gynecological surveillance efficacy of currently used guidelines, we plan to carry out a nationwide, prospective study.

## 5. Conclusions

In conclusion, we assessed gynecological tumors and gynecological management in 505 LS carriers diagnosed in our center. Of 164 LS carriers, data on gynecological surveillance and/or risk-reducing surgery were available and evaluated. Risk-reducing surgery was eventually performed in nearly half of these 164 LS carriers, especially *MSH6* carriers. Most of the surveillance visits were performed in time. Three out of eight carriers with endometrial hyperplasia had gynecological complaints. Four of seven EC patients and the one patient with OC had gynecological complaints, whereas three ECs were found in asymptomatic women. All ECs were diagnosed in FIGO I. This questions whether, and if so, to what extent, female LS carriers may benefit from current gynecological surveillance programs. Larger, prospective studies are warranted to determine the efficacy and compliance of gynecological surveillance strategies in LS carriers. Additionally, alternatives should be assessed: for example, gynecological surveillance in which female carriers only reach out when having gynecological complaints. As a consequence, uniform surveillance strategies can be developed and implemented.

## Figures and Tables

**Figure 1 cancers-13-00459-f001:**
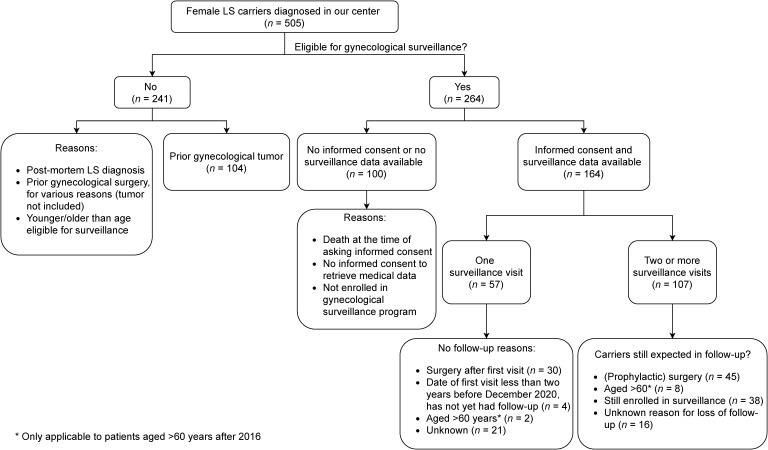
Overview of female Lynch syndrome (LS) carriers.

**Table 1 cancers-13-00459-t001:** Characteristics of carriers with gynecological surveillance.

Characteristics	Total	*MLH1*	*MSH2*	*MSH6*	*PMS2*	*p*-Value
All patients, *N*	164	38	25	82	19	
Of which diagnosed before 2016	139	33	23	70	13	
Of which diagnosed since 2016	25	5	2	12	6	
Age at first visit, before 2016, median (IQR, range)	46.0 yr (IQR 37.9–53.6, range 21.5–75.0)	44.5 yr (IQR 36.2–52.2, range 21.5–65.2)	43.4 yr (IQR 34.7–50.8, range 26.8–54.6)	48.2 yr (IQR 39.2–55.6, range 28.6–75.0)	49.2 yr (IQR 34.9–56.5, range 26.3–70.9)	0.143
Age at first visit since 2016, median (IQR, range)	53.8 yr (IQR 42.4–61.1, range 30.0–71.3)	48.3 yr (IQR 37.1–51.1, range 30.0–61.1)	57.4 yr (IQR 54.7–60.2, range 54.7–60.2)	47.6 yr (IQR 42.3–63.0, range 38.5–71.3)	58.4 yr (IQR 56.5–64.7, range 53.7–65.3)	0.175
Follow-up years	685.4 yr	237.7 yr	146.2 yr	238.3 yr	63.2 yr	
Number of follow-up years, median (IQR)	5.6 yr (IQR 3.0–9.0)	6.3 yr (IQR 5.2–10.2)	7.0 yr (IQR 4.5–10.4)	4.2 yr (IQR 2.4–7.7)	3.7 yr (IQR 2.0–7.6)	0.009
Number of visits ^Δ^, median N (IQR)	3 (IQR 2–6)	4 (IQR 2–8)	3 (IQR 2–6)	3 (IQR 1–5)	3 (IQR 1–5)	<0.001
Visits with tests performed, N (% of total visits)						
Transvaginal ultrasound	522 (76.8%)	184 (79.0%)	79 (72.5%)	207 (77.8%)	52 (72.2%)	0.431
Ca125	276 (40.6%)	128 (54.9%)	18 (16.5%)	117 (44.0%)	13 (18.1%)	<0.001
Endometrial sampling	244 (35.9%)	68 (29.2%)	34 (31.2%)	99 (37.2%)	43 (59.7%)	<0.001

^Δ^ Including first surveillance visit; Yr = years, visits = surveillance visits

**Table 2 cancers-13-00459-t002:** Characteristics of gynecological surveillance visits.

Characteristics	Total	*MLH1*	*MSH2*	*MSH6*	*PMS2*	*p*-Value
All patients, *N*	164	38	25	82	19	
Patients with only one surveillance visit, *N*	57	8	7	37	5	0.040
Surgery after first visit, including for tumor	30	3	2	21 ^‡^	4	
Recent primary visit (<2 years)	5	0	0	3	1	
Aged >60 years (since 2016)	2	1	0	1	0	
Reason unknown	20	4	5	12	0	
Abnormalities at first surveillance visit, *N* **						
Ca125 > 35 kU/L	2	1	0	1	0	
Thickened Endometrium *	5	2	0	3	0	
Hyperplasia	5	0	0	4	1	
EC	1	0	0	1 ^‡^	0	
Patients with ≥2 surveillance visits, *N* (expected count ^∆^)	107 (127)	30 (34)	18 (23)	45 (57)	14 (14)	
Interval between visits before 2016, median (IQR)	1.0 (IQR 0.9–1.5)	1.0 (IQR 0.8–1.3)	1.1 (IQR 1.0–2.0)	1.0 (IQR 0.6–1.4)	1.1 (IQR 1.0–1.4)	0.001
Interval between visits since 2016, median (IQR)	1.0 (IQR 1.0–1.2)	1.0 (IQR 1.0–1.3)	1.1 (IQR 1.0–1.2)	1.0 (IQR 0.8–1.1)	1.0 (IQR 1.0–1.1)	0.277
Subsequent surveillance visit <24 months before 2016, *N* of total (%)	343/417 (82.3%)	131/155 (84.5%)	47/64 (73.43%)	139/167 (83.2%)	26/31 (83.9%)	0.249
Subsequent surveillance visit <15 months/all visits since 2016, *N* of total (%)	80/96 (83.3%)	26/37 (72.2%)	11/14 (78.6%)	24/26 (92.3%)	18/19 (94.7%)	0.091
Abnormalities at ≥2 surveillance visit, *N* (% of carriers with ≥2 surveillance visits) **						
Ca125 > 35 kU/L	8 (7.5%)	2 (6.7%)	1 (5.6%)	2 (4.4%)	1 (7.1%)	
Thickened Endometrium *	10 (9.3%)	0	2 (11.1%)	6 (13.3%)	2 (14.3%)	
Hyperplasia	3	1	1	1	0	
Tumor	6	2	3	1	0	
EC	5	2	2	1	0	
OC	1	0	1	0	0	

* If endometrial thickness was >10 mm in premenopausal women (during proliferative phase) or >4 mm in postmenopausal women, it was considered to be thickened. ** Carriers with abnormalities identified upon more than one visit were only counted once (only the first visit with an abnormal outcome). ^‡^ One endometrial cancer (EC) was identified in an *MSH6* carrier, with subsequent surgery (more information in Table 4). ^∆^ Expected count was based on the identified number of patients with two or more surveillance visits summed with the number of patients with unknown cause for cessation of follow-up.

**Table 3 cancers-13-00459-t003:** Gynecological hyperplasia and malignancies identified during surveillance.

Tumor (FIGO, Grade) or Hyperplasia	MMR Gene Involved	Age	Surveillance Visits Prior to Hyperplasia or Tumor, *N*	Years to Previous Visit	Tests Performed at Previous Surveillance Visit	Complaints Prior to Identification of Hyperplasia or Tumor
Endometrial hyperplasia	*MSH6*	46	N.A.	N.A.	N.A.	Hypermenorrhea
Endometrial hyperplasia	*MSH6*	46	N.A.	N.A.	N.A.	Unknown
Endometrial hyperplasia	*MSH6*	57	N.A.	N.A.	N.A.	Unknown
Endometrial hyperplasia	*MSH6*	51	N.A.	N.A.	N.A.	Unknown
Endometrial hyperplasia	*PMS2*	49	N.A.	N.A.	N.A.	Irregular cycle
Endometrial hyperplasia	*MSH6*	52	5	N.A.	N.A.	Hypermenorrhea, hyperplasia identified upon pathological examination after risk-reducing surgery, preoperative endometrial sampling could not rule out hyperplasia
Endometrial hyperplasia	*MSH2*	38	4	1.0	Transvaginal ultrasound, CA125 (9 kU/L), endometrial sampling (outcome: no premalignancy)	No complaints
Endometrial hyperplasia	*MLH1*	48	3	1.1	Transvaginal ultrasound	No complaints
EC (FIGO IA grade 1)	*MSH6*	64	N.A.	N.A.	N.A.	No complaints
EC (FIGO IB, grade 1)	*MSH6*	58	1	N.A.	N.A.	No complaints, EC identified upon pathological examination after risk-reducing surgery, preoperative normal endometrial sampling
EC (FIGO IA grade 1)	*MLH1*	54	4	1.1	Transvaginal ultrasound, endometrial sampling (outcome: weakly proliferating endometrium)	No complaints, no anomalies at ultrasound, but endometrial sampling suggestive of pre-malignancy.
EC (FIGO I)	*MLH1*	52	1	0.5	Transvaginal ultrasound	Complaints: irregular blood loss
EC (FIGO IA, grade 2)	*MSH2*	54	2	1.5	Transvaginal ultrasound	Complaints: postmenopausal blood loss
OC (FIGO IV)	*MSH2*	48	1	1.0	Transvaginal ultrasound unknown, no Ca125 or endometrial sampling	Complaints: intermenstrual blood loss, menorrhagia.
EC (FIGO IA, grade 1)	*MSH2*	37	2	1.5	Transvaginal ultrasound, Ca125 (11 kU/L)	Complaints: blood loss after miscarriage
EC (FIGO IA grade 2)	*MSH6*	59	3	0.1	Transvaginal ultrasound, Ca125 (outcome unknown), no endometrial sampling	Complaints: postmenopausal blood loss since last visit

**Table 4 cancers-13-00459-t004:** Gynecological surgery in patients at gynecological surveillance.

Characteristics	Total	*MLH1*	*MSH2*	*MSH6*	*PMS2*	*p*-Value
Gynecological surgery ^‡^, *N* (% of total carriers)	73 (45.5%)	14 (36.8%)	7 (32.0%)	44 (53.7%)	8 (42.1%)	0.015
Of which performed before 2016	54	7	6	35	6	0.004
Of which performed since 2016	19	7	1	9	2	1.00
Type of surgery						
Hysterectomy	11	3	3	5	0	
Ovariectomy ^∆^	4	1	2	1	0	
Hysterectomy with bilateral salpingo oophorectomy	58	10	2	38	8	
Age at surgery before 2016, median (IQR, range)	51 years (IQR 45–54, range 20–72)	52 years (IQR 42–53, range 35–69)	45 years (IQR 38–46, range 38–48)	51 years (IQR 46–56, range 20–72)	53 years (IQR 50–61, range 35–62)	0.076
Age at surgery since 2016, median (IQR, range)	51 years (IQR 47–56, range 42–65)	49 years (IQR 47–51, range 46–56)	60 years ^∫^	51 years (IQR 43–52, range 42–61)	64 years (IQR 62–65, range 62–65)	0.082
Surveillance visits prior to surgery, median *N* (IQR)	3 (IQR 1–5)	4 (IQR 2–7)	2 (IQR 1–4)	3 (IQR 1–5)	2 (IQR 1–3)	0.002
Reasons for surgery						0.158
Risk-reducing surgery, *N* (% of total per gene)	53 (71.6%)	10 (71.4%)	3 (37.5%)	34 (77.3%)	6 (75.0%)	
Surgery after abnormal surveillance visit, *N* (% of total per gene)	13 (17.6%)	1 (7.1%)	2 (25.0%)	8 (18.2%)	2 (2.7%)	
Other (% of total per gene)	3 (4.1%)	2 (14.2%)	1 (12.5%)	0	0	
Unknown (% per gene)	5 (6.8%)	1 (7.1%)	2 (25.0%)	2 (4.5%)	0	
Abnormalities ^†^ identified upon pathological assessment, *N* (% of total per gene)	3 (4.1%)	0	0	3 ^†^ (4.1%)	0	

^‡^ Surgery due to identification of a gynecological malignancy is herein not included. † Hyperplasia (*n* = 2) and EC (*n* = 1) were identified in pathology specimens of *MSH6* carriers with surgery. ^∆^ Ovariectomy with or without fallopian tubes ^∫^
*N* = 1, therefore no interquartile range (IQR) or range.

## Data Availability

The data presented in this study are available on request from the corresponding author. The data are not publicly available due to ethical reasons (data being registry data).

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
