# Peer review of "Gynecological Surveillance and Surgery Outcomes in Dutch Lynch Syndrome Carriers"

_cancers, 2021, doi:10.3390/cancers13030459_

Round 1
Reviewer 1 Report
This a an informative and predictive study which could be beneficial for the LS carriers. One of the concerns is about the sample size and very narrow gepgraphical distribution of the study participants data. A broader spectrum of data compilation would strengthen the findings and the manuscript. Here are some of the comments:
- It will be informative to mention if there is any evidence that the germline variants in DNA mismatch repair particularly causes these two or three cancer types indicated or there are other cancers studied in this context.
-
In line 112, it would be appropriate to include an explanation of CA125 measurement. Also the FIGO could be explained.
-
In Table 1, row 4 onwards, the columns look overcrowded, may be decreasing font or removing repeated words like 'years' to yr might make it present better to understand.
- It is unclear whether there were cases where multiple MMR genes were involved for cancer initiation?
Author Response
This a an informative and predictive study which could be beneficial for the LS carriers. One of the concerns is about the sample size and very narrow gepgraphical distribution of the study participants data. A broader spectrum of data compilation would strengthen the findings and the manuscript.
We want to thank the reviewers for their thorough and critical appraisal of our manuscript. We share the concerns of reviewer 1 regarding the sample size, however, due to the retrospective nature of this study, it was not possible to assess gynecological outcomes in more Lynch syndrome carriers. Nevertheless, until now, our cohort is one of the largest describing gynecological surveillance outcomes in Lynch syndrome carriers. In order to more accurately assess outcomes of gynecological surveillance visits in a larger cohort, we plan to carry out a national, prospective cohort study.
Here are some of the comments:
- It will be informative to mention if there is any evidence that the germline variants in DNA mismatch repair particularly causes these two or three cancer types indicated or there are other cancers studied in this context.
Pathogenic variants in DNA mismatch repair genes are known to be causative of among others colorectal cancer, endometrial cancer, and ovarian cancer. LS carriers are particularly prone to develop these cancers, however, they are also at risk –albeit lower- to develop other tumors, such as tumors of the small intestine, brain, and skin. This information was added to the introduction (line 62), with a supporting reference.
- In line 112, it would be appropriate to include an explanation of CA125 measurement. Also the FIGO could be explained.
In line 124, we clarified that CA125 is a tumor marker, that may be elevated in blood of patients with OC or EC. We also included a reference supporting this explanation. Additionally, we explained in line 108 that FIGO stages are used as a classification for gynecological tumors.
- In Table 1, row 4 onwards, the columns look overcrowded, may be decreasing font or removing repeated words like 'years' to yr might make it present better to understand.
In Table 1, the fond was decreased to 8 and the number of words in the columns was decreased: ‘years’ was abbreviated to ‘yr’, ‘surveillance visits’ was abbreviated to ‘visits’, and ‘in carriers diagnosed before or since 2016’ was changed to ‘before 2016’ or ‘since 2016’. Abbreviations were explained beneath this table.
- It is unclear whether there were cases where multiple MMR genes were involved for cancer initiation?
In this study, gynecological surveillance outcomes were only assessed in LS carriers with one pathogenic MMR germline variant. Carriers with biallelic pathogenic MMR germline variants, having constitutional mismatch repair deficiency (CMMRD) syndrome, usually develop brain tumors or colorectal cancer before adulthood and were therefore excluded for further analyses. There were no patients in our center with pathogenic variants in more than one MMR gene. This was added to the materials and methods section (line 102).
Reviewer 2 Report
This is an interesting retrospective study concerning gynecological screening in women with Lynch syndrome. The value of the surveillance appears to be low and this is broadly in line with previous experience with cancer screening in many other high-risk populations.
The paper is well written and the analysis is appropriate. I would recommend adding an information how the women with LS (and their gynecologists) are advised about the screening. Do they get a written material for their gynecologist oultining the risks and the recommended schedule?
Author Response
This is an interesting retrospective study concerning gynecological screening in women with Lynch syndrome. The value of the surveillance appears to be low and this is broadly in line with previous experience with cancer screening in many other high-risk populations.
The paper is well written and the analysis is appropriate. I would recommend adding an information how the women with LS (and their gynecologists) are advised about the screening. Do they get a written material for their gynecologist oultining the risks and the recommended schedule?
We thank the reviewer for the kind remarks. Upon LS diagnosis in our center, the genetic counselor informs female LS carriers about their cancer risks, among others gynecological tumor risks, and options for gynecological surveillance and/or surgery. In case carriers have (nearly) reached age 40, they are referred to a gynecologist for further information regarding surveillance and/or surgery, including their advantages and disadvantages. Carriers referred to the department of gynecology in our center additionally receive a brochure summarizing tumor risks and the effects of gynecological surveillance and surgery. This was added to the materials and methods section (line 113).